# Mapping Evidence Regarding Decision-Making on Contraceptive Use among Adolescents in Sub-Saharan Africa: A Scoping Review

**DOI:** 10.3390/ijerph20032744

**Published:** 2023-02-03

**Authors:** Mumbi Chola, Khumbulani W. Hlongwana, Themba G. Ginindza

**Affiliations:** 1Discipline of Public Health Medicine, School of Nursing and Public Health, University of KwaZulu-Natal, Durban 4041, South Africa; 2Department of Epidemiology and Biostatistics, School of Public Health, University of Zambia, Lusaka 10101, Zambia; 3Cancer & Infectious Diseases Epidemiology Research Unit (CIDERU), College of Health Sciences, University of KwaZulu-Natal, Durban 4041, South Africa

**Keywords:** adolescents, decision-making, contraception, contraceptive use, Sub-Saharan Africa

## Abstract

This scoping review mapped and synthesised existing evidence on the influence of individual, parental, peer, and societal-related factors on adolescents’ decisions to use contraception in sub-Saharan Africa (SSA). Peer-reviewed and review articles published before May 2022, targeting adolescents aged 10–19 years were searched in PubMed, MEDLINE with Full Text via EBSCOhost, PsychINFO via EBSCOhost, CINAHL with Full Text via EBSCOhost, Google Scholar, Science Direct, and Scopus databases. Seven studies were included and analysed using thematic analysis based on the social-ecological model (SEM) and reported using the preferred reporting items for systematic reviews and meta-analyses (PRISMA). Individual (fear of side effects, fear of infertility), parental (parental disappointment and disapproval), peer (social stigma), partner (association with promiscuity and multiple sexual partners), societal and community (contraceptive use disapproval and stigma), and institutional and environmental factors (lack of privacy and confidentiality) influence contraceptive decisions among adolescents. These also include a lack of accurate information, social exclusion, negative health provider attitudes, and a lack of infrastructure that provides privacy and safe spaces. Identifying and addressing core issues within the context of local cultural practices that restrict contraceptive use is important. Holistic, inclusive approaches that promote the well-being of adolescents must be utilised to provide a conducive environment that ensures privacy, confidentiality, safety, and easy access to contraceptive services.

## 1. Introduction

The World Health Organisation (WHO) defines family planning as a voluntary and informed choice that allows individuals and couples to decide how many and when to have their children. This is attained through contraception, which is the act of preventing pregnancy through the use of a device, medication, procedure, or change in behaviour [1]. This involves the use of contraceptive methods [2], including modern contraceptive methods, which are products designed to prevent pregnancy resulting from sexual intercourse [3], such as implants, female sterilisation, male sterilisation, intrauterine devices, condoms, oral contraceptives, emergency contraceptives, and other contraceptives, such as diaphragms, spermicides, and vaginal rings [3].

The prevalence of contraceptive use among women globally varies. According to statistics, in 2015, two out of every three married women or those in a union aged 15 to 49 years used a modern or traditional contraceptive [4]. The lowest numbers were reported in Eastern and Southern Africa (38.6%) and West and Central Africa (17.6%) [4]. A study conducted in 29 countries in sub-Saharan Africa (SSA) found that only 24.7% of adolescent girls and young women aged 15–24 years use modern contraception, with the highest and lowest prevalence reported in Lesotho (59.2%) and Chad (5.1%), respectively [5]. Compared to all women, contraceptive prevalence rates were much lower among adolescents [6,7,8], with only 15% of girls aged 15–19 years who were either married or in a union using modern contraceptives [9]. Adolescents include those aged between 10 and 19 years [10]. They can be further divided into early adolescents (11–13 years), adolescents (14–17 years), and young adults (18–25 years), who include those aged 18–19 years [11].

Low rates of contraception use expose adolescents to the risk of adolescent pregnancy, which remains a serious public health problem, especially in Africa. Statistics from low- and middle-income countries show that for adolescents aged 15–19 years, 50% of the approximately 21 million pregnancies recorded annually are unplanned and result in approximately 12 million births [12]. Adolescent pregnancy is associated with adverse health outcomes in this age group, compared to older women, including higher risks for early neonatal death, anaemia, puerperal endometritis, operative vaginal delivery, and episiotomy [13]. They also face a significantly increased risk of maternal mortality, obstructed labour, and obstetric fistula, which reduces their chances of getting an education or being employed [14,15,16]. Additionally, the children born to these teenage mothers have an increased chance of death, low birth weight, small-for-gestational-age infants, and dropping out of school compared to their peers [13,17].

Factors that are associated with low contraceptive use among adolescents were reported in various studies [18,19,20,21,22,23,24,25]. However, it is particularly important to understand what influences adolescents’ decisions on whether or not to use contraceptives. It is vital to understand the factors that influence adolescents, positively or negatively, as they decide whether to use contraceptives. The decision to use or not to use contraceptives is largely driven by various influences [18,19,20,21,22,23,24,25]. These include individual or intrapersonal influences [18,19,23,24,25], partner influences [19,20,26], peer influences [18,22,24], parental [22,24,25], and societal influences [18,20,22,25]. These include factors such as fear of side effects, fear of infertility at the individual level, parental disappointment and disapproval at the parental level, social stigma at the peer level, association with promiscuity and multiple sexual partners at the partner level and the societal level, as well as societal and cultural norms that disapprove of and stigmatise contraceptive use [18,19,20,21,22,23,24,25].

This information will be useful for policymakers and program managers as they seek to address low contraceptive use in this age group and prevent pregnancy complications. It will also be important for improving and re-shaping health policies and programmes targeted at adolescent sexual and reproductive health.

This scoping review, therefore, was aimed at mapping current literature on what influences decision-making among adolescent girls regarding contraceptive use, especially parental factors, partner factors, societal factors, and peer-related factors, and how they influence adolescent girls’ contraceptive decisions.

## 2. Methods and Materials

### 2.1. Design

This was a scoping review of published peer-reviewed literature on factors that influence adolescent girls’ decision-making regarding contraceptive use in sub-Saharan Africa (SSA). This review is nested in a larger study, which aimed to examine the levels, patterns, and trends of contraception use among adolescent girls and understand the influencers and motivators of their contraceptive decisions. While the protocol for this review was previously published [27], this scoping review included studies published before May 2022, when the literature search was conducted. The review was guided by the preferred reporting items for systematic review and meta-analysis protocols (PRISMA-P) (Figure 1) [28] and was based on Arksey and O’Malley’s methodological framework for scoping studies [29].

### 2.2. Identifying the Research Question

The research question that guided this scoping review was: what is the available evidence on the individual, parental, peer, societal, and cultural factors that influence decision-making in contraceptive use among adolescents? In answering this research question, the review was guided by the socio-ecological model (SEM).

### 2.3. Eligibility Criteria

Eligible studies that presented findings on adolescent girls aged 10–19 years; included decision-making in contraceptive use; influences of parents, peers, partners, and society on adolescent girls’ contraceptive decisions, and individual or “self” influences on adolescents’ decisions to use contraceptives, and focused on populations based in SSA that were included in the review. The eligibility of the research question was determined using the population—concept—context (PCC) framework (Table 1) as recommended by the Joanna Briggs Institute for scoping reviews [30].

### 2.4. Identifying Relevant Studies (Search Strategy)

The databases that were searched for articles meeting the eligibility criteria included PubMed, MEDLINE with Full Text via EBSCOhost, PsychINFO via EBSCOhost, CINAHL with Full Text via EBSCOhost, Google Scholar, Science Direct, and Scopus. The search adhered to the PRISMA guidelines [32]. Studies published in SSA before May 2022 were included in the search. The search also included eligible studies listed in the citations of the selected studies. The keywords used to search the databases were: adolescent, girls, contraceptive use, and decision-making. The Boolean term “AND” was used to separate keywords. Medical subject headings (MeSH) terms were included in the keyword search. Only articles published in English were included in the study.

A library was created for this review using EndNote 20 referencing software. The primary investigator conducted a comprehensive search and screening of the study titles from the above-mentioned databases. All studies with eligible titles were exported to the EndNote library, and all duplicates were removed before the abstract screening. Two reviewers independently conducted abstract screening followed by full-article screening of selected studies, using standardised tools, with guidance from the eligibility criteria. Disputes were discussed and resolved based on the eligibility criteria. To optimise the article search strategy, library services at the University of KwaZulu-Natal were utilised to help with retrieving and finding articles to be included in the full-article screening. Reporting on these was conducted using the preferred reporting items for systematic reviews and meta-analyses (PRISMA) chart [32], shown in Figure 1.

### 2.5. Data Extraction/Charting

Table 2 shows the data extraction form that was developed, piloted, and used to extract and process the relevant information from each of the studies included in the review. All variables that were included in the form were aimed at answering the research question. Data extraction was conducted by the primary author and was reviewed by the other authors. The authors reviewed, discussed all discrepancies, and agreed on the final interpretation. A unique code was assigned to all reviewed articles to assist with keeping track of all reviewed articles and those excluded during the data charting process.

### 2.6. Collating, Summarising, and Reporting the Results

To improve quality, an iterative process was used to continuously review the data. The data were analysed using thematic content analysis. The manuscripts included in the analysis were coded around the following themes: individual, partner, peer, parental, and societal influences on the decision to use contraceptives. These themes are based on McLeroy and colleagues’ social-ecological model (SEM) of health promotion [33]. According to their model, health behaviour and promotion are interrelated and take place across multiple levels at the individual, interpersonal, institutional, community, and policy levels. The model [33] argues that patterned behaviour is determined by the following:(1)Intrapersonal factors, which are individual characteristics.(2)Interpersonal factors, which are processes and principal groups that can include social networks (both formal and informal), and social support systems, such as families, and friendship networks.(3)Institutional factors, including social establishments that have structural characteristics and prescribed (and informal) formalities on how they operate.(4)Community factors, which include relationships between organisations, institutions, and informal networks within specified borders.(5)Public policy, which can include local, and national laws, regulations, and policies.

Various studies across different health topics used this model to determine, understand, and describe health issues and patterns that are complex in nature [34,35]. This multidimensional outlook is vital to understanding and explaining the different factors that influence decision-making regarding contraceptive use. Individual level or intrapersonal factors are factors such as lack of or limited knowledge, misaligned concerns, religious beliefs, and misinformation about the safety and effectiveness of contraceptives; interpersonal factors included partner, peer, parental, and societal influences or any interactions with social networks (both formal and informal) and social support systems; institutional level factors included characteristics and health system activities that affect the use of contraceptives; and community-level factors, which included interactions between and within organisations in the health system, community, and other social networks that influence contraceptive use [36]. Policy-level factors are government policies that affect contraceptive use [36]. Based on the themes identified, a narrative was written describing the findings.

Overall, 160 articles were identified based on the title screening search criteria, after screening through 68,041 titles (see Table 3). At the title screening stage, 67,881 articles were removed because they met the exclusion criteria (i.e., those with no evidence on adolescent girls aged 10–19, contraceptive use, decision-making, and those conducted outside of SSA). A total of 32 duplicates were removed, and 128 titles were included for abstract screening. These articles were exported to the Endnote library for abstract screening. Following the screening of the abstracts, 14 were included for full article screening, while 114 were excluded. A total of seven articles were excluded because four focused on the utilisation of contraceptives; one on decision-making for sexual and reproductive health (SRH); one on acceptance of contraceptives; and one on perceptions about contraceptives. Only seven articles met the eligibility criteria and were included in the final analysis and qualitative synthesis.

### 2.7. Characteristics of Studies Included

Table 4 describes the details of the studies that were included in this review. All studies were published before May 2022. All seven studies [18,19,20,22,23,24,25] were qualitative. Two studies were conducted in Malawi [18,20], two in Nigeria [23,24], and one in Ghana [19], Kenya [22], and South Africa [25], respectively. The total sample size of the included studies was 511 participants, the majority of whom were females.

## 3. Results

Contraceptive decisions made by adolescent girls were influenced by various factors and at different levels. These included individual, parental, peer, partner, societal, community, and institutional and environmental influences, which either positively or negatively influenced adolescent girls’ contraceptive decisions, particularly non-barrier, hormonal contraceptives. The findings below are described based on the SEM.

### 3.1. Intrapersonal or Individual-Level Factors

Individual-level or intrapersonal factors that influence the decision to use contraceptives among adolescent girls were explored in all seven studies [18,19,20,22,23,24,25]. These studies found that, at the individual level, various factors influence the decision to use contraceptives [19].

### 3.2. Health Concerns

Across all studies, fear of side effects was a major factor in influencing contraceptive decisions. Reported side effects included missing menses and heavy bleeding [20], changes in weight and menstrual cycles, stomach pain, and reduced libido [22]. They also reported ‘‘waist pains’’ following sex, loss of weight, menstrual irregularities, and prolonged absence of menstruation [25]. This was mainly linked to hormonal contraceptives and their reported side effects, which deterred adolescent girls, particularly those who were unmarried and nulliparous, from using them. Adolescent girls in Malawi highlighted a lack of information due to health providers not giving detailed information to adolescents as a factor, while poor knowledge about the various types of hormonal contraceptives and how they work was reported in Ghana [19]. Adolescents in Ghana could name at least one hormonal contraceptive but lacked information on how they work.

### 3.3. Fertility Concerns

Fear of infertility was also reported among adolescents in Nigeria [23,24]. This was mainly among unmarried women who expressed concerns about having difficulties conceiving once they became married [23]. Peer pressure, encouragement from male sexual partners [25], lack of efficacy to organise hormonal contraceptives [19], and preservation of fertility [24], based on alleged effects that contraceptives have on future fertility, were all factors in the decision to use contraceptives. Fear of the effects of contraception on infertility or future fertility among adolescent girls and community members was a common concern in almost all the studies [18,20,22,23,24,25].

### 3.4. Stigma Concerns

Adolescent girls in Ghana feared disclosing the use of hormonal contraceptives for fear of being labelled as ‘bad girls’ [19]. In Malawi, adolescent girls’ fears and concerns revolved around misinformation about hormonal contraception, such as the fact that it causes cancer [20]. In Kenya, the fears centred around the shame of dishonouring parents, the stigma from their peers and community, and the social withdrawal that is associated with teenage pregnancy [22]. The stigma towards contraceptive use in their communities was mainly among adults and their peers who related the use of contraceptives with sexual promiscuity and lustful behaviour among girls [22]. Adolescents in South Africa reported fearing being condemned [25]. There were also negative attitudes towards the use of hormonal contraceptives among adolescents, and fear of disclosure of contraceptive use also influenced adolescent girls’ decision to use contraceptives [19]. Beliefs that contraceptives were for married women also influenced adolescent girls’ decisions to use them [20].

Another factor and motivator for contraceptive use among adolescents in Nigeria was the pursuit of education [24], while in Kenya, the consequences of adolescent pregnancy on future educational attainment, which they viewed as leading to poverty, was also a factor in their contraceptive decisions [22].

### 3.5. Interpersonal Factors

#### 3.5.1. Parental Factors

There was an influence from parents, both positive and negative, on adolescents’ decisions to use contraceptives, as reported in Kenya, Malawi, Nigeria, Ghana, and South Africa, both positively and negatively. On the negative side, adolescent girls in Malawi reported that parents and guardians had negative attitudes toward the use of contraceptives. Parents do not talk about contraceptives and their advantages and disadvantages. They view sex and contraceptives as immoral and do not discuss these issues because of their cultural and religious beliefs [20]. Similar views were expressed by adolescents in Ghana. Adolescents feared that their parents would view them as bad girls because they are not supposed to be having sex, which is viewed as immoral behaviour and a sign of disrespect to their parents [19]. This would result in their parents being disappointed in them and possibly chasing them away from home or disowning them [19,20].

Positive influences were also reported. Adolescents in South Africa reported that their mothers imposed contraceptives on them once their menses started [25], while in Kenya, they reported that their mothers advised or directed them to use condoms and not contraceptives amid concerns about their effects on the health and future fertility of the girls [22]. Some adolescents mentioned that conversations with their mothers helped them make decisions regarding contraceptive use. Adolescents in Ghana were more positive about using hormonal contraceptives, especially when discussing it with their mothers first before choosing a particular method. This gave them some assurance that their mothers would be fine with them using the contraceptives if they knew about it [19]. In Nigeria, conversations with mothers were also instrumental in positively influencing decisions on contraceptive use [24].

#### 3.5.2. Peer Factors

Peers also have an influence, both positive and negative, on adolescent girls’ decisions to use contraceptives. Adolescents in Malawi reported that conversations they had with their peers about infertility and social norms related to contraceptive use reinforced their decision not to use contraceptives, particularly among unmarried, nulliparous adolescents [18]. The implied association, by peers, between contraceptive use and promiscuous behaviour also discouraged contraceptive use [18]. Positive peer influences were reported among adolescents in Nigeria. In Nigeria, however, adolescent girls reported using contraceptives because this was what their peers were doing [24]. The girls would turn to their peers, and those they surrounded themselves with, to inform their decisions.

#### 3.5.3. Partner Factors

Sexual partners have both a positive and negative influence on the contraceptive decisions made by adolescent girls. They either encouraged or discouraged contraceptive use. Some partners positively influence contraceptive decisions among adolescent girls, although the motive is for their benefit. In Kenya, adolescent girls mentioned that they decided to use contraceptives because their partners did not want to endure the embarrassment of impregnating a young girl [22], while others did it because their partners wanted to stop using condoms and have unprotected sex. Where sexual partners negatively influenced adolescent girls’ decisions to use contraceptives and discouraged contraceptive use, it was based on their concerns regarding the effect that contraceptives may have on the future fertility of the girls [22]. Adolescents also made decisions not to use contraceptives due to their partners’ concerns that using contraceptives signified having multiple sexual partners, which discouraged their use [18]. Adolescents in Nigeria, however, reported that sexual partners played a role in deciding which contraceptive method to use, particularly among married adolescents [24]. Other issues, such as the duration of the relationship, also influenced contraception decisions among adolescents [20]. Adolescents who were in long-term relationships, compared to those in short-term relationships, were more likely to choose to use contraceptives [20]. However, the definition of short-term and long-term relationships was not detailed in the articles reviewed [25].

#### 3.5.4. Societal, Cultural and Community Factors

Fear of the effects of contraception on infertility or future fertility among adolescent girls and community members was a common concern in almost all the studies [18,20,22,23,24,25]. Myths about the link between hormonal, non-barrier contraceptives and infertility were widespread, as were fears that using contraceptives causes infertility. It was believed that these methods were to be used by married women or for child spacing [18]. In Kenya, perceptions concerning not being able to give birth to a healthy baby in the future were a major influence on contraceptive decisions among adolescent girls [22]. Women are under social and cultural pressure to prove their fertility and reproduce, especially young women and newlyweds. The threat that infertility places on their future marriage prospects was also a major concern and influenced decisions about contraceptive use [24]. Societal and cultural norms that disapprove of and stigmatise premarital sex and contraceptive use were also key influencers on decisions to use contraceptives [19]. The perception that adolescents who use contraceptives are promiscuous also influenced decisions on contraceptive use [18,22]. Other issues, such as the duration of the relationship, also influenced contraception decisions among adolescents [20]. The fears also centred around the shame of dishonouring parents, the stigma from their peers and community, and the social withdrawal that is associated with teenage pregnancy [22]. Adolescents in South Africa also reported fearing being condemned [25].

#### 3.5.5. Institutional and Environmental Factors

Institutional and environmental factors relating to accessing contraceptives and how they influence decisions to use them were reported among adolescents in Ghana [19], Malawi [20], and South Africa [25]. The absence of privacy in health facilities, and the lack of a conducive environment and policies that hinder access to contraceptives all influence contraception decisions among adolescents in Malawi [20]. In South Africa, the attitudes of healthcare workers, particularly nurses, influenced decisions to use contraception among adolescent girls [25]. Adolescents cited harsh treatment and scolding by nurses as negative influences. They also reported that nurses often attempted to coerce the girls into using injectables, as they were seen as the most reliable contraceptives [25].

## 4. Discussion

This is the first scoping review to map and synthesize existing evidence on the influence of individual, parental, peer, and societal factors on adolescents’ contraceptive decisions in Sub-Saharan Africa. The results of this study indicate that various individual, parental, peer, and partner, societal and community, and institutional and environmental factors influence, both positively and negatively, adolescents’ decisions on whether to use contraceptives. However, some of the main concerns under these influences were common and cross-cutting and are discussed below.

### 4.1. Intrapersonal Factors Influencing Adolescent Girls’ Contraceptive Decisions

The fear of the side effects arising from contraceptive use was also reported across all the studies reviewed. Similar studies among adolescents and women, in general, reported comparable findings [37,38,39,40,41]. The fear of side effects is a major hindrance to contraceptive use, thus influencing contraceptive decisions among adolescent girls and women at large. This is due to the consequences that these side effects have on their lives, such as the strain and threats they place on relationships [42], including threats of physical violence [25].

Another major deterrent to contraceptive use was the alleged side effect of it causing sterility. Fears about contraceptive use causing infertility and affecting future fertility were a major influence on decision-making regarding using contraceptives among adolescents in all the studies reviewed [18,19,20,22,23,24,25]. This was explored across all levels in this study, i.e., individual, peer and partner, community, and societal levels. Other studies also reported on this belief that using modern contraceptives can cause infertility [39,40,43,44,45,46,47]. Women believed that infertility caused by contraception was caused by blood accumulation and blockage in the womb [25,41,48,49], damage to or spoiling the womb [44,49], displacement and internal movement of the contraceptive device (particularly the implants and IUD) causing damage to the body organs or going missing in the body [19,40,50], toxicity and contamination of the blood [23,48], and loss of libido, leading to inability to conceive [25,51]. Most women in African societies face enormous pressure to have children, and having children and being a mother are viewed positively and perceived as giving women elevated social status and respect in their communities [52]. As such, adolescent girls and women who cannot bear children may experience challenges socially and culturally, including disruption of their relationships or marriages, boyfriends or husbands resorting to polygamy, divorce, or promiscuity by the boyfriend or husband [41,45]. Therefore, where their ability to reproduce is threatened or is seemingly affected by contraceptive use, adolescent girls and women, in general, will opt to not use contraceptives or discontinue use where they started.

The lack of accurate information about contraceptives could be contributing to the fear of side effects. Studies showed that the information adolescents had about contraceptives, how they work, and how to use them was often inadequate, incomplete, and sometimes wrong [20,23,25], and studies reported similar findings [53]. Lack of information also influences contraceptive decisions, and it was cited as a deterrent to using contraceptives [20,23,41,54]. Adolescent girls also lacked credible sources of information, relying on friends, peers, and parents [18,19,22,24,25]. This lack of accurate information about contraceptives and how they work and the lack of reliable sources for accurate information can result in misconceptions and misinformation. When the fear of infertility, lack of information, and reliable sources are coupled with the high value placed on fertility in most sub-Saharan African cultures, it places great pressure on women to bear children, which contributes to adolescents deciding not to use contraceptives. Having a child accrues certain advantages and status in society, not just to the woman, but to the entire family [48]. Therefore, the fear of infertility and future fertility being affected results in women not using or discontinuing the use of contraceptives, irrespective of marital status, to safeguard their future fertility. The inability to bear children can have drastic consequences for the woman, both socially and culturally, which may include marital disruption, divorce, promiscuity by the husband, or even polygamy [41,45].

### 4.2. Interpersonal Factors Influencing Adolescent Girls’ Contraceptive Decisions

#### 4.2.1. Parental Factors

As found in other studies, parents, particularly mothers, have a significant influence on the contraceptive decisions of adolescent girls, whether positively or negatively. Their discomfort with discussing sex and sex-related issues with their children [55] can negatively influence contraceptive decisions among adolescent girls. The absence of parental communication on sexuality and sex-related issues with adolescents provides an opportunity for external influencers, such as peers, partners, and the media. Regarding positive parental influences on adolescent contraceptive decisions, studies revealed that conversations between children and their parents are effective at communicating sexual and reproductive information and instilling values in the children [56,57]. As a result of the enormous influence that parents have on adolescent girls’ social, emotional, and cognitive development, they must be more involved in ensuring that the girls’ contraceptive decisions and sexual and reproductive health needs are met [58].

#### 4.2.2. Partner Factors

Sexual partners have a substantial influence on the contraceptive decisions adolescent girls make. Adolescent girls are the principal users of contraceptives and typically bear responsibility for contraceptive use, despite their partners having significant influence in deciding the type of contraceptive used [41]. The demand for their partners’ consent, the denial of their partner’s approval, and the adolescent girls’ inability to seek consent from their partners all negatively affect and influence the contraceptive decisions that the adolescent girls make [59].

However, studies also show the positive influences that partners have on contraceptive decisions. Communication with partners about contraceptive use positively influences the use of contraceptives among adolescent girls and young women, particularly with non-barrier methods [60]. For married adolescents, this is consistent with research findings on spousal communication and decision-making that are husband-centred [61,62,63], which align with family planning interventions that target male involvement, focusing mostly on married couples and husbands [64].

#### 4.2.3. Peer Factors

Peer relationships are a major influence on contraceptive use among adolescent girls. As parental and community influence on adolescents diminishes, peer relationships start to take on a more significant role in adolescents’ lives. The views of peers can negatively influence contraceptive decisions among adolescents and prevent them from using contraceptive services [55]. Peer relationships can also positively influence contraceptive decisions. Conversations about contraceptives with peers were positively associated with the use of non-barrier contraceptives, more so among those who are single, whether they have a child or not [60].

### 4.3. Society and Community Factors Influencing Adolescent Girls’ Contraceptive Decisions

Studies in this review reported the belief that contraceptives are for married women and those who had children before—not single, nulliparous women. Comparable findings were reported in other studies [65]. This belief and the fear of stigma, social exclusion, and the linking of contraceptive use with promiscuous behaviour contribute to the contraceptive decisions that adolescents make, which lead to the non-use of contraceptives. They fear that if their use of contraceptives became common knowledge, they would be labelled as being promiscuous, ‘bad girls’ with no good morals, or accused of prostitution not only by peers, but also by their parents [18,19,22]. This was also reported in other studies [41,65,66,67]. This could result in social exclusion and their being the centre of gossip among their friends [18,19]. Among married adolescents, they feared that even discussing the use of contraceptives with their partners might imply having multiple sexual partners [18]. With such consequences that would affect their family and social lives, adolescents are inclined to decide against using contraceptives or discontinue their use to avoid the stress that comes with these consequences.

### 4.4. Institutional Factors Influencing Adolescent Girls’ Contraceptive Decisions

Challenges with access and the availability of a conducive environment were also raised. Health facilities and communities also do not provide a conducive, judgement-free environment in which adolescents can exercise their freedom and make independent decisions [20]. Among the issues raised were lack of privacy, ease of access, youth-friendly services, and services targeting older adolescents. The negative attitude of healthcare staff also discouraged accessing contraceptives from health facilities [54,66,67,68]. Healthcare workers’ negative and paternalistic attitudes towards adolescents’ use of contraceptives were documented in various studies [41,66,69]. They discourage adolescent girls from using contraceptives, holding the view that they should be used by married adults, as they are largely meant for child spacing. Adolescents view communication with nurses as being below standard, incomplete, and often unilateral, with the nurses using technical language and not being in sync with the needs of adolescents [41]. Challenges raised by adolescents in the studies reviewed included being lectured about premarital sex, harassment, being scolded, and being denied services until they answered a barrage of questions. Another concern raised by adolescent girls was that their parents and community members, whom they did not want to know about their contraceptive use, also used the same health facilities. Adolescents, therefore, preferred to access services in other locations, such as drug stores.

## 5. Conclusions

Our review revealed various individual, parental, peer, and partner, societal and community, and institutional and environmental factors that influence adolescent girls’ contraceptive decisions. However, these factors were common and cross-cutting across these levels. The main factors were the fear of the side effects, particularly infertility and future fertility, lack of accurate information, beliefs, and cultural norms about who can use contraceptives, which contributed to the fear of stigma and social exclusion from friends and society. Other concerns included the impact on relationships and marriages, as well as the connection between the use of contraceptives and promiscuity. Challenges with access to contraceptives, particularly from health facilities, as a result of negative attitudes among health providers and a lack of infrastructure that provides privacy and safe spaces, were another major concern. These barriers remain a major challenge in most African countries and will need to be addressed if contraceptive use among adolescents is to improve.

Identifying and addressing the core issues within the context of local cultural norms is essential. To improve demand, access, and usage of contraceptives among adolescent girls, the implementation of policies and programmes that target this age group must be re-strategised. Ensuring constant engagement to enhance acceptance of contraceptives at the individual and community level, particularly among adolescents and young adults, is essential. To help dispel fears and misconceptions about contraception, communities and intended users should be given complete and accurate information. Accessing information from reliable sources, particularly health providers, should also be actively encouraged. Youth-friendly spaces should be the norm in health facilities. Training health providers and having a younger staff who can identify and communicate with adolescents in a respectful, nonjudgmental manner can help change health staff attitudes toward young people. Infrastructure-wise, locating youth-friendly spaces in more secluded places that afford privacy and confidentiality can help improve access to and uptake of contraceptive services in this age group. In these efforts to address these challenges, it is imperative to adopt holistic and inclusive approaches that embrace broader stakeholder involvement, utilise evidence-based data, and promote the well-being of adolescents.

## Figures and Tables

**Figure 1 ijerph-20-02744-f001:**
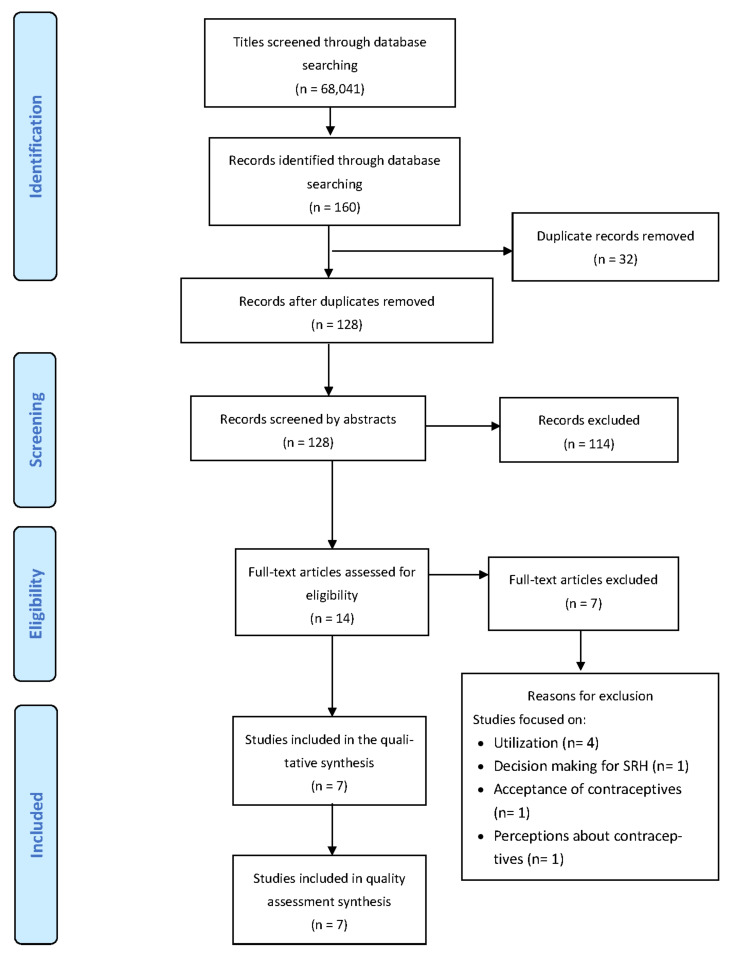
PRISMA 2009 flow diagram.

**Table 1 ijerph-20-02744-t001:** PCC Framework.

Population	Adolescents aged 10–19 years—as defined by the World Health Organisation (WHO) [31].
Concept	Decision-making in contraceptive use. This includes factors that adolescents consider and processes that they go through in deciding whether or not to use contraceptives.
Context	Sub-Saharan Africa—including studies from sub-Saharan Africa.

**Table 2 ijerph-20-02744-t002:** Data extraction form.

**Author and Date**
**Study Title**
**Study Population** Gender
**Methodology** Geographical setting (country) Study site Study type Study design
**Data Collection Methods** Data collection tools Data collection method
**Sampling** Sample size
**Intervention (Contraception)** Contraceptive use Decision-making
**Data Analysis** Data analysis type Data analysis method
**Results** Most important finding Other findings
**Conclusion**

**Table 3 ijerph-20-02744-t003:** Title screening search results.

No.	Search Engine	Search Date	Search Terms	Search Results	Final Selected Articles
1	PubMed	19 April 2022	(((Adolescent) AND (girls)) AND (contraceptive use)) AND (decision-making)	1706 titles	44 articles
2	Google Scholar	21 April 2022	adolescent AND girls AND contraceptive use AND decision-making	64,400 titles	86 articles
3	Science Direct	27 April 2022	adolescent AND girls AND contraceptive use AND decision-making	1731 titles	11 titles
4	EBSCOhost-MEDLINE with Full Text	27 April 2022	Adolescent AND girls AND contraceptive use AND decision-making	112 titles	12 articles
5	EBSCOhost-APA PsychINFO	27 April 2022			
6	EBSCOhost-Academic Search Complete	27 April 2022			
7	Scopus	27 April 2022	(TITLE-ABS-KEY (adolescent) AND TITLE-ABS-KEY (girls) AND TITLE-ABS-KEY (contraceptive AND use) AND TITLE-ABS-KEY (decision AND making))	92 titles	7 articles
			TOTAL	68,041 titles	160 titles

**Table 4 ijerph-20-02744-t004:** Characteristics of included studies.

**Author and Year**	**Study Objective**	**Study Setting (Country)**	**Study Design**	**Study Population Sample Size**	**Data Collection**
Bhushan, N.L., et al., (2021) [18]	This study qualitatively looked at the nature of contraceptive conversations among AGYW who were enrolled in a sexual and reproductive health study conducted in Lilongwe, Malawi called Girl Power. The study sought to understand the context, content, and impact of contraceptive-related conversations between AGYW and their sexual partners, peers, and older women in their families.	Malawi	This was a Qualitative study conducted as part of Girl Power-Malawi, a quasi-experimental study that implemented across four health facilities in Lilongwe, Malawi.	Included 60 AGYW aged 15–24 years	60 Individual in-depth interviews were conducted.
Boamah-Kaali, E.A., et al., (2021) [19]	This was a study conducted in the Kintampo area of Ghana which aimed at exploring, from the adolescents’ perspective, factors that limit the uptake of hormonal contraceptives.	Ghana	An exploratory study using qualitative data collection methods.	Included 38 adolescent girls aged 15–19 years.	2 focus group discussions and 16 in-depth interviews.
Dombola, G. M., et al., (2021) [20]	This study aimed to understand contraceptive use and decision-making among young adolescents aged between 10 and 14 years.	Malawi	Qualitative based design	Included 26 young adolescents aged 10–14 years.	2 focus group discussions and 26 in-depth interviews.
Harrington, E. K., et al., (2021) [22]	The objective of this study was to explore how adolescent girls and young women aged 15–19 years in Kenya viewed their contraceptive needs and also how, within their social contexts, they make decisions to use contraceptives. They also studied social influences on decisions to use contraceptives among adolescents who were at risk for pregnancy. They sought to provide nuanced insights into the contraceptive behaviours of adolescents.	Kenya	Qualitative study	Included 86 adolescent girls aged 15–19 years	40 IDIs and 6 FGDs were conducted.
Otoide, V.O., et al., (2001) [23]	This study was conducted in Nigeria and sought to explore the beliefs and attitudes regarding abortion and also explore the attitudes and beliefs of adolescents concerning contraceptive use. The study also explored the fears that adolescent girls have about the availability, perceived advantages, side effects, and reasons for the use or non-use of contraceptives.	Nigeria	Qualitative study	Included 149 AGYW aged 15–24 years	20 focus-group sessions. The number of participants per group ranged from 6 to 10.
Sanchez, E.K., et al., (2020) [24]	This was a qualitative study that aimed to determine what and who influences contraceptive-seeking behaviours among adolescent girls in Nigeria.	Nigeria	This was a qualitative study that was conducted as part of a larger study investigating the sustainability and impact of the Nigerian Urban Reproductive Health Initiative (NURHI)	AGYW aged 15–24 Years with a total of 117 participants	12 focus group discussions (FGD) with three Nigerians with each group comprising 8–12 participants.
Wood, K. and R. Jewkes (2006) [25]	This study was conducted in Limpopo province, South Africa, to collect information that could be used to enhance adolescent women’s access to contraceptives as well as the quality of contraceptive services generally. The scope of inquiry included the circumstances of and influences on girls’ contraceptive-seeking practices and decision-making. The investigation also focused on the circumstances as well as the factors that influenced adolescent girls’ decisions about using contraceptives.	South Africa	Qualitative study	Adolescent girls aged 14–20 years with a total of 35 participants,	35 individual, semi-structured interviews and 5 group discussions. Each group comprised between 3 and 6 informants.

## Data Availability

All data generated from this study will be included in the published scoping review article and will also be available on request.

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
