# Peer review of "Mapping Evidence Regarding Decision-Making on Contraceptive Use among Adolescents in Sub-Saharan Africa: A Scoping Review"

_ijerph, 2023, doi:10.3390/ijerph20032744_

Round 1
Reviewer 1 Report
Thank you for the opportunity to review this article. This paper addresses an important aspect of understanding contraceptive decision-making among sub-Saharan African adolescents by reviewing the literature. This paper is not yet ready for publication and needs significant proofreading to achieve proficient English-language readability. The results and discussion should be reworked for clarity of each SEM level findings and the significance of results in relation to the literature.
Line 31 – “stigma” or “stigmatisation”, not “stigmatise”.
Lines 33-35 – this sentence restates the previous parentheticals.
Line 37 – unnecessary comma
Line 45 – “which is attained through” –consider combining this idea with the next sentence
Line 46 – “disrupt reproduction” – consider “prevent pregnancy”
Line 47 - “copulation” – consider “sexual intercourse”
Line 51 – It’s not the “prevalence of women” that varies; it’s either the proportion of women varies or the prevalence of contraception use varies among women
Line 52 – “a union” – do you mean in a relationship? Clarify.
Line 55 – “statistics” weren’t lower – say “rates” were lower
Line 61 – “low rates of…” “to the risk of adolescent pregnancy” (extra “is” before “remains”)
Lines 64 has “maternal death”, line 66 restates “maternal mortality” – same thing
Line 68-69 – focus this sentence only on the children, there is erroneous repeat of ”adolescent mothers also face higher risks of” that can be taken out
Line 83- “…useful to vital for…” – this paper needs proofreading.
Due to the large amount of grammatical and wording errors, the rest of my review will just refer to sections overall rather than specific lines.
Figure 1 – error under “reasons for exclusion” – cannot read last line.
Methods:
Define SSA in the manuscript text (not just the abstract) before using the acronym. Define PRISMA too.
The research question should specify the SSA population. Too many research sub-questions, unnecessary to break them out into four numbers. This could easily be one combined sentence about societal, peer, parental, and individual factors.
Eligibility Criteria – too much about the pro/cons of different methods – just talk about what you used and why you used it, not what you didn’t use.
How did you identify the relevant studies related to SSA from the initial number? Did you manually go through to select those involving SSA only? (Later you mention the title screening – this should be mentioned earlier as your method of screening.)
“To improve quality, the data were extracted and continuously reviewed” – what does this mean? Excerpts? Quotes? Codes? Do you mean iteratively review?
Lines 194-205: Unnecessary to include a synopsis of each level in this form. Particularly as Lines 206-218 then redefine the same things in paragraph form.
Search queries: Why does the pubmed query have all those extra parentheses, when the others do not? Why not include alternatives phrases to “decision making”?
Table 4 – Make the text alignment “Left”, not “Justify”/full line, it makes the columns hard to read. Make sentence style uniform: some begin with “XX interviews” and others “they conducted XX focus groups”
Results
254 – What does “waist pains” mean? Abdominal pain? Pelvic pain?
Consider breaking up these large “Individual Level” paragraphs into subheadings, such as “Fertility Concerns”, “Health Concerns”, “Stigma Concerns”. There is some repetition of similar fears in multiple places in these paragraphs – try to consolidate.
Some of the individual “social” concerns sound more like “peer factors” in the interpersonal section – consolidate and focus so that the different SEM levels are not replicating the same point more than once.
Partners – Lines 325-326 I think you mean they “either” encouraged or discouraged, not “both”. Should state “some/many/most” positively influence rather than “they”
Similar issues in other categories with overlap between multiple SEM levels and typographical errors (i.e. line 360 – the letter “n”)
Discussion
Is it the first paper to review this question targeting SSA? If so, say so.
Separating out the SEM levels into the discussion is unnecessary and has led to significant repetition in each subsection. Instead, restate very briefly the significance of the results, then move on to interpretation.
Major repetition of results between Line 383-392 – make the discussion your place to interpret these results (such as the 392-393 sentence), not to restate results. There is more restating and little interpretation for the rest of the paragraph. If citing outside literature in relation to your data, state what you are talking about and whether they either agree or disagree and why.
408 – How do you know the fear stems from lack of accurate information? As the second half of the paragraph states, the fear may be due to the perceived severity of the consequences of loss of fertility, and thus may not be responsive to providing more accurate information as an intervention.
471 – does “contraceptives” here include condoms? Were there different views about condom use than medical contraceptives? If so, explain.
Conclusion – should not be introducing new information. Usually you include these future directions in the discussion, and use conclusion to provide a take home point.
Abbreviations – missing SSA, and PRISMA is not what PRISMA stands for.
Sources: Some URLs begin with your host institution’s library search, such as line 607 “https://ukzn.idm.oclc.org/login?url=https://search.ebscohost.com/” – make sure that the link is the DOI or Pubmed.
Several articles are cited as “[Internet]” when they should be cited as articles, such as #21 and #39 and #52, 54, 56, 61. 62.
Author Response
Responses to comments from Reviewer 1
Reviewer Comment 1: Thank you for the opportunity to review this article. This paper addresses an important aspect of understanding contraceptive decision-making among sub-Saharan African adolescents by reviewing the literature. This paper is not yet ready for publication and needs significant proofreading to achieve proficient English-language readability. The results and discussion should be reworked for clarity of each SEM level findings and the significance of results in relation to the literature.
Response 1: Thank you for taking the time out of your busy schedule to review our manuscript. We will do our best to address the comments provided. We appreciate all the feedback provided as we endeavour to improve our manuscript and share the findings from our research.
Reviewer Comment 2: Line 31 – “stigma” or “stigmatisation”, not “stigmatise”.
Response 2: Line 31: This has been revised based on the suggestion provided.
Reviewer Comment 3: Lines 33-35 – this sentence restates the previous parentheticals.
Response 3: This has been revised to include only those that are not in parentheses.
Reviewer Comment 4:Line 37 – unnecessary comma
Response 4: This has been deleted.
Reviewer Comment 5: Line 45 – “which is attained through” –consider combining this idea with the next sentence
Response 5: This has been revised based on the suggestion provided.
Reviewer Comment 6: Line 46 – “disrupt reproduction” – consider “prevent pregnancy”
Response 6: The suggested changes have been made
Reviewer Comment 7: Line 47 - “copulation” – consider “sexual intercourse”
Response 7: The suggested changes have been made
Reviewer Comment 8: Line 51 – It’s not the “prevalence of women” that varies; it’s either the proportion of women varies or the prevalence of contraception use varies among women
Response 8: This has been revised based on the recommendation from the reviewer.
Reviewer Comment 9: Line 52 – “a union” – do you mean in a relationship? Clarify.
Response 9: Yes. This refers to women who are in a relationship.
Reviewer Comment 10: Line 55 – “statistics” weren’t lower – say “rates” were lower
Response 10: This has been revised to improve clarity.
Reviewer Comment 11: Line 61 – “low rates of…” “to the risk of adolescent pregnancy” (extra “is” before “remains”)
Response 11: This has been revised as suggested.
Reviewer Comment 12: Lines 64 has “maternal death”, line 66 restates “maternal mortality” – same thing
Response 12: This has been revised. maternal death has been removed since maternal mortality is referred to in the following sentence.
Reviewer Comment 13: Line 68-69 – focus this sentence only on the children, there is erroneous repeat of ”adolescent mothers also face higher risks of” that can be taken out
Response 13: This has been revised as suggested.
Reviewer Comment 14: Line 83- “…useful to vital for…” – this paper needs proofreading.
Response 14: This has been revised.
Reviewer Comment 15: Due to the large amount of grammatical and wording errors, the rest of my review will just refer to sections overall rather than specific lines.
Response 15: The manuscript has been checked and edited for grammatical and wording errors which have been addressed.
Reviewer Comment 16: Figure 1 – error under “reasons for exclusion” – cannot read last line.
Response 16: The size of the box has been adjusted to ensure all the text is visible.
17 Methods:
Reviewer Comment 18: Define SSA in the manuscript text (not just the abstract) before using the acronym. Define PRISMA too.
Response 18: Both SSA and PRISMA have been defined
Reviewer Comment 19: The research question should specify the SSA population. Too many research sub-questions, unnecessary to break them out into four numbers. This could easily be one combined sentence about societal, peer, parental, and individual factors.
Response 19: The sub-research questions have been collapsed into one research question as suggested.
Reviewer Comment 20: Eligibility Criteria – too much about the pro/cons of different methods – just talk about what you used and why you used it, not what you didn’t use.
Response 20: The comparison of the PICO and PCC frameworks has been removed and only the reference to the method we used has been included.
Reviewer Comment 21: How did you identify the relevant studies related to SSA from the initial number? Did you manually go through to select those involving SSA only? (Later you mention the title screening – this should be mentioned earlier as your method of screening.)
Response 21: Lines 127-129 explain how the screening was done from title screening to abstract and full article screening. Referring to Table 2: data extraction form, we included the geographical setting/country which helped identify studies cone in SSA.
Reviewer Comment 22: “To improve quality, the data were extracted and continuously reviewed” – what does this mean? Excerpts? Quotes? Codes? Do you mean iteratively review?
Response 22: What we mean here is iteratively reviewed. This has been revised to improve clarity.
Reviewer Comment 23: Lines 194-205: Unnecessary to include a synopsis of each level in this form. Particularly as Lines 206-218 then redefine the same things in paragraph form.
Response 23: The synopsis in lines 206 – 216 was added to highlight some examples of the different levels which we felt were important as we describe the SEM model and some of the issues we would focus on in the review. In the paragraph, we describe how other studies have used the model and what they have found as factors that influence decision making.
Reviewer Comment 24: Search queries: Why does the pubmed query have all those extra parentheses, when the others do not? Why not include alternatives phrases to “decision making”?
Response 24: All the search terms in the table were copied as they appeared from the search results for each search engine. PubMed had the extra parentheses while the others did not. The focus of this paper was on decision making. We will consider including alternatives in future studies of a similar nature.
Reviewer Comment 25: Table 4 – Make the text alignment “Left”, not “Justify”/full line, it makes the columns hard to read. Make sentence style uniform: some begin with “XX interviews” and others “they conducted XX focus groups”
Response 25: The text alignment has been changed based on the suggestion. The sentence style form has been changed and is now uniform.
26 Results
Reviewer Comment 27: 254 – What does “waist pains” mean? Abdominal pain? Pelvic pain?
Response 27: This was a finding in one of the studies that was included in the review. There was no indication of whether “waist pains” meant abdominal or pelvic pain. Therefore, we quoted it as it was from the study by Wood and Jewkes
Reviewer Comment 28: Consider breaking up these large “Individual Level” paragraphs into subheadings, such as “Fertility Concerns”, “Health Concerns”, “Stigma Concerns”. There is some repetition of similar fears in multiple places in these paragraphs – try to consolidate.
Response 28: The individual-level section has been restructured with the proposed subheadings included.
Reviewer Comment 29: Some of the individual “social” concerns sound more like “peer factors” in the interpersonal section – consolidate and focus so that the different SEM levels are not replicating the same point more than once.
Response 29: Individual concerns that were more of peer factors have been realigned to reduce repetitions. This has been done across all SEM levels to ensure that there is minimal repetition.
Reviewer Comment 30: Partners – Lines 325-326 I think you mean they “either” encouraged or discouraged, not “both”. Should state “some/many/most” positively influence rather than “they”
Response 30: This has been revised as suggested.
Reviewer Comment 31: Similar issues in other categories with overlap between multiple SEM levels and typographical errors (i.e. line 360 – the letter “n”)
Response 31: This typographical error has been addressed.
32 Discussion
Reviewer Comment 33: Is it the first paper to review this question targeting SSA? If so, say so.
Response 33: Thank you. We have mentioned that it is the first review to look at this subject in SSA.
Reviewer Comment 34: Separating out the SEM levels into the discussion is unnecessary and has led to significant repetition in each subsection. Instead, restate very briefly the significance of the results, then move on to interpretation.
Response 34: The repetition has been addressed. However, we maintained the structure of the discussion to align with the SEM because we wanted to highlight and discuss the findings in line with the different subsections. We believe this helps to highlight literature that specifically looks at these subsections.
Reviewer Comment 35: Major repetition of results between Line 383-392 – make the discussion your place to interpret these results (such as the 392-393 sentence), not to restate results. There is more restating and little interpretation for the rest of the paragraph. If citing outside literature in relation to your data, state what you are talking about and whether they either agree or disagree and why.
Response 35: The repetition of results has been removed and more text discussing the findings has been added.
Reviewer Comment 36: 408 – How do you know the fear stems from lack of accurate information? As the second half of the paragraph states, the fear may be due to the perceived severity of the consequences of loss of fertility, and thus may not be responsive to providing more accurate information as an intervention.
Response 36: This has been rephrased to state that fear of side effects could be contributing to this.
Reviewer Comment 37: 471 – does “contraceptives” here include condoms? Were there different views about condom use than medical contraceptives? If so, explain.
Reviewer Comment 37: This refers primarily to hormonal contraceptives such as injectables and pills. This has also been clarified in the manuscript.
Reviewer Comment 38: Conclusion – should not be introducing new information. Usually you include these future directions in the discussion, and use conclusion to provide a take home point.
Response 38: We have removed the information relating to the Zambia Adolescent Health strategy. Information that has been retained related to some recommendations on how some of the challenges identified in this review can be addressed.
Reviewer Comment 39: Abbreviations – missing SSA, and PRISMA is not what PRISMA stands for.
Response 39: SSA has been added and PRISMA has been corrected.
Reviewer Comment 40: Sources: Some URLs begin with your host institution’s library search, such as line 607 “https://ukzn.idm.oclc.org/login?url=https://search.ebscohost.com/” – make sure that the link is the DOI or Pubmed.
Response 40: All the references have been edited to remove the host institution from the URL.
Reviewer Comment 41: Several articles are cited as “[Internet]” when they should be cited as articles, such as #21 and #39 and #52, 54, 56, 61. 62.
Response 41: This has been corrected. All the references have been cited as articles.

Reviewer 2 Report
Thank you for the opportunity to review this important and timely scoping review. It was very interesting.
Abstract: The abstract mentions the use of PRISMA and PRISMA-ScR ( see line 28), but the body of the text only refers to PRISMA. I would recommend to either change in body of text or change what is said in abstract. I was actually interested in what the difference would be between the two and didn't see that, only PRISMA was mentioned in text.
Introduction: The paragraph starting on line 51, refers to prevalence but seems rushed and this reviewer is left with questions concerning whether the authors know what the prevalence of adolescence contraception use is specifically in SubSaharan Africa as a grouping since it is a mix of developed and underdeveloped countries. There is no definition of underdeveloped country as well.
-The explanation for why this review is needed or why it is important is weakly stated. The authors state it is a serious public health problem but there is no supporting documentation as to these statements, no statistics, no data, or specific complications.
-It is recommended that a definition of what exactly is meant by contraception use and what low versus high would be. This is could be a broad and general definition because it will vary depending on which types of contraception methods are being studied. But something should be said about the authors definition of contraception use and low versus high. Lines 65-70 then are mixed up and confusing, there seems to be a mix of children factors and maternal factors. It is recommended that a slow and thoughtful grouping be done in this section.
Line 72 should have a reference.
Lines 83-86-recommended that the authors rework as to the why and complete the sentence on line 86. Is this a review as to explain the why this review is important from a policy and health program perspective or are you just suggesting potential importance among what should be a long list of importance for this review.
Methods and Materials: Prisma Flow Diagram- Reasons for the exclusion given in the flow diagram given for full-text exclusions but not others but it is mentioned in the text. The reviewer is still unclear why acceptance and perception articles were not included, might want to say more about those exclusions.
Results: Line 320-it is recommended that the authors describe or give examples of what used contraceptives are.
Line 332, 386 need spell and sentence checks.
Line 337-It is recommended that the authors explain short-term and long term relationships or mention that in the articles reviewed there was no definition.
line 457- incomplete sentence, missing a word
Conclusion: Line 518- Sentences need to be reworked.
Would recommend a spell check of the entire manuscript for some minor edits. Lastly, are the SSA countries all universal healthcare since there was no discussion of costs as a factor or concern.
The reviewer would like to thank the authors for an opportunity to review this very interesting piece. It should be fine with some minor revisions and work in a few sections.
Author Response
Responses to comments from Reviewer 2
Reviewer Comment 1: Abstract: The abstract mentions the use of PRISMA and PRISMA-ScR (see line 28), but the body of the text only refers to PRISMA. I would recommend to either change in body of text or change what is said in abstract. I was actually interested in what the difference would be between the two and didn't see that, only PRISMA was mentioned in text.
Response 1: This has been revised to include only PRISMA since this is what was focused on in the review.
Reviewer Comment 2: Introduction: The paragraph starting on line 51, refers to prevalence but seems rushed and this reviewer is left with questions concerning whether the authors know what the prevalence of adolescence contraception use is specifically in Sub Saharan Africa as a grouping since it is a mix of developed and underdeveloped countries. There is no definition of underdeveloped country as well.
Response 2: The section has been revised with some statistics on contraceptive use in SSA countries included.
Reviewer Comment 3: The explanation for why this review is needed or why it is important is weakly stated. The authors state it is a serious public health problem but there is no supporting documentation as to these statements, no statistics, no data, or specific complications.
Response 3: We have added statistics on adolescent pregnancy to highlight the extent of the problem which results in part from low contraceptive use.
Reviewer Comment 4: It is recommended that a definition of what exactly is meant by contraception use and what low versus high would be. This is could be a broad and general definition because it will vary depending on which types of contraception methods are being studied. But something should be said about the authors definition of contraception use and low versus high. Lines 65-70 then are mixed up and confusing, there seems to be a mix of children factors and maternal factors. It is recommended that a slow and thoughtful grouping be done in this section.
Response 4: A definition of contraception has been included in the first paragraph of the introduction just after the definition of family planning. The distinction between health risks to the mother and the children born to adolescent mothers has been made. Lines 62-67 refer to the mothers while lines 67-70 refer to the children.
Reviewer Comment 5: Line 72 should have a reference.
Response 5: References have been added.
Reviewer Comment 6: Lines 83-86- recommended that the authors rework as to the why and complete the sentence on line 86. Is this a review as to explain the why this review is important from a policy and health program perspective or are you just suggesting potential importance among what should be a long list of importance for this review.
Response 6: We are suggesting the potential importance of this review. We do, however, recognise that this is a long list which we cannot exhaustively list in this manuscript.
Reviewer Comment 7: Methods and Materials: Prisma Flow Diagram- Reasons for the exclusion given in the flow diagram given for full-text exclusions but not others but it is mentioned in the text. The reviewer is still unclear why acceptance and perception articles were not included, might want to say more about those exclusions.
Response 7: Acceptance and perception studies were not included because they did meet the eligibility criteria. The focus of this review was on factors that influence decision making hence the exclusion of articles on acceptance and perception.
Reviewer Comment 8: Results: Line 320-it is recommended that the authors describe or give examples of what used contraceptives are.
Response 8: The article cited refers only to modern contraceptives but does not detail the specific contraceptives
Reviewer Comment 9: Line 332, 386 need spell and sentence checks.
Response 9: Spelling and sentence checks have been done for both sentences have been corrected
Reviewer Comment 10: Line 337-It is recommended that the authors explain short-term and long term relationships or mention that in the articles reviewed there was no definition.
Response 10: The definition of what constituted long and short term relationships was not defined in the reviewed articles. A sentence explaining this has been added.
Reviewer Comment 11: line 457- incomplete sentence, missing a word
Response 11: The sentence has been revised to improve clarity.
Reviewer Comment 12: Conclusion: Line 518- Sentences need to be reworked.
Response 12: The sentence has been revised to improve clarity.
Reviewer Comment 13: Would recommend a spell check of the entire manuscript for some minor edits. Lastly, are the SSA countries all universal healthcare since there was no discussion of costs as a factor or concern.
Response 13: The manuscript has been run through a spell checker to address all spelling and grammatical errors.
Reviewer Comment 14: The reviewer would like to thank the authors for an opportunity to review this very interesting piece. It should be fine with some minor revisions and work in a few sections.
Response 14: Thank you for taking the time to review our manuscript and provide feedback. We truly appreciate your efforts in helping us improve our manuscript and helping us publish our work.

Round 2
Reviewer 1 Report
Good edits. Thank you for bringing light to an important understudied area of this topic.